# Charge Transport Enhancement in BiVO_4_ Photoanode for Efficient Solar Water Oxidation

**DOI:** 10.3390/ma16093414

**Published:** 2023-04-27

**Authors:** Zhidong Li, Zhibin Xie, Weibang Li, Hafiz Sartaj Aziz, Muhammad Abbas, Zhuanghao Zheng, Zhenghua Su, Ping Fan, Shuo Chen, Guangxing Liang

**Affiliations:** Shenzhen Key Laboratory of Advanced Thin Films and Applications, Key Laboratory of Optoelectronic Devices and Systems of Ministry of Education and Guangdong Province, College of Physics and Optoelectronic Engineering, Shenzhen University, Shenzhen 518060, China

**Keywords:** BiVO_4_, photoanode, electro-deposition, water oxidation, catalyst

## Abstract

Photoelectrochemical (PEC) water splitting in a pH-neutral electrolyte has attracted more and more attention in the field of sustainable energy. Bismuth vanadate (BiVO_4_) is a highly promising photoanode material for PEC water splitting. Additionally, cobaltous phosphate (CoPi) is a material that can be synthesized from Earth’s rich materials and operates stably in pH-neutral conditions. Herein, we propose a strategy to enhance the charge transport ability and improve PEC performance by electrodepositing the in situ synthesis of a CoPi layer on the BiVO_4_. With the CoPi co-catalyst, the water oxidation reaction can be accelerated and charge recombination centers are effectively passivated on BiVO_4_. The BiVO_4_/CoPi photoanode shows a significantly enhanced photocurrent density (*J*_ph_) and applied bias photon-to-current efficiency (ABPE), which are 1.8 and 3.2 times higher than those of a single BiVO_4_ layer, respectively. Finally, the FTO/BiVO_4_/CoPi photoanode displays a photocurrent density of 1.39 mA cm^−2^ at 1.23 V_RHE_, an onset potential (*V*_on_) of 0.30 V_RHE_, and an ABPE of 0.45%, paving a potential path for future hydrogen evolution by solar-driven water splitting.

## 1. Introduction

The production of hydrogen through solar-driven water splitting typically stores solar energy in chemical bonds, which is a potential strategy for overcoming shortages of energy and related global environmental problems [1,2,3,4,5]. The characteristics of photoanode materials decide the solar-to-hydrogen (STH) efficiency of a photoelectrochemical (PEC) device, because the kinetic demand of oxygen evolution reaction (OER) in the PEC water-splitting procedure is higher than the hydrogen evolution reaction (HER) [6,7,8]. To date, various neutral electrolytes have been studied to function as PEC photoanodes, such as TiO_2_ [9,10], Fe_2_O_3_ [11], WO_3_ [12], and BiVO_4_ [13,14,15,16,17,18]. Among them, the monoclinic bismuth vanadate (BiVO_4_) has received significant attention because of its favorable band-edge positions, its moderate band gap energy (*E*_g_, 2.4–2.5 eV) [19], low cost, etc. However, the surface charge recombination and severe bulk defect delay the OER kinetics limits PEC performance [20]. BiVO_4_ photoanodes have been the subject of numerous studies aimed at enhancing their PEC performance, including doping [21], building heterojunction [22], crystal facet or morphology engineering [23], oxygen vacancies (O_v_) introduction, and the surface modification of oxygen evolution catalysts (OECs), etc. [24,25].

The surface modification of oxygen evolution catalysts is a more promising method for enhancing the PEC performance of BiVO_4_ photoanodes than its other counterparts, due to the effective passivation of the surface charge recombination centers and improving the interfacial OER kinetics for PEC water oxidation. For instance, J. Hu et al. successfully synthesized iron oxhydroxide (FeOOH) with different crystalline phases (α-, β-, and δ-) through a regulated solvothermal pathway, where the electrocatalytic OER activity of β-FeOOH was highest [26]. Carbon quantum dots (CQDs) are also able to remarkably improve electrocatalytic OER activity owing to the increased O_v_ density; at the same time, they can form heterojunctions with semiconductors, thereby effectively promoting charge separation and transport [27]. Moreover, L. Wu et al. synthesized the nickel boride (NiB) layer by adjusting the composition of the neutral electrolyte. The borates with B-O bonds become promoters of catalyst activity by accelerating proton-coupled electron transfer and interact with Ni^2+^ ions to inhibit charge recombination on BiVO_4_ surface, thereby reporting a *J*_ph_ of 6.0 mA cm^−2^ at 1.23 V_RHE_ [14]. Cobaltous phosphate (CoPi) is an effective electrocatalyst for water oxidation and was first reported by G. Nocera and W. Kanan, who also demonstrated that hydrogen phosphate ions are proton acceptors in oxygen production reactions under neutral pH conditions [28]. Moreover, J. Durrant et al. discussed the oxidation degree of the CoPi catalyst on the BiVO_4_ photoanode under simulated sunlight irradiation and determined the appropriate degree of catalyst oxidation to drive substantial water oxidation. Additionally, the relative kinetics of water oxidation on the surfaces of electrocatalyst and semiconductor and the kinetics of holes transfer to electrocatalysts were discussed for the first time [18].

In this work, a BiVO_4_ thin film was initially fabricated, which is inherently beneficial for charge transport through the adjustment the electrodeposition time of the BiOI precursor film and the dropping of excessive vanadium source solution for annealing in a muffle furnace. Then, the cobaltous phosphate (CoPi) layer was electrodeposited on optimal BiVO_4_ electrodes while the composition of neutral electrolyte was regulated by adding cobaltous nitrate and phosphate species. Finally, the FTO/BiVO_4_/CoPi photoanode was successfully prepared. Combining the electrocatalytic and photoelectric technologies, the CoPi for OER catalytic activity was investigated in detail and notable results were obtained, e.g., a *J*_ph_ of 1.39 mA cm^−2^ (at 1.23 V_RHE_) and an undoped ABPE of 0.45% under AM 1.5 G illumination.

## 2. Materials and Methods

### 2.1. Preparation of BiOI Precursor Film

The BiOI precursor film was electrodeposited on an FTO glass substrate with an effective area of 1.5 cm × 2 cm in a three-electrode system, where a Pt-foil was used as the counter electrode and an Ag/AgCl with saturated KCl solution was used as the reference electrode. The BiOI precursor deposition solution was fabricated by dissolving lactic acid (0.03 M), KI (0.4 M), and Bi(NO_3_)_3_·5H_2_O (0.015 M) in deionized water (100 mL), with 1,4-Benzoquinone (0.046 M) in ethanol (40 mL) solution. The pH of the mixed solution was adjusted to 3.7 by adding 0.1 M nitric acid aqueous solution after stirring for 20 min. Initially, a 60 s deposition was conducted at −0.40 V_Ag/AgCl_ to prevent the falling off of BiOI films from the surface of the FTO substrate. After the initial deposition, the BiOI film was obtained at a constant voltage of −0.25 V_Ag/AgCl_ with different deposition durations, then rinsed thoroughly with deionized water and dried in a drying oven.

### 2.2. Preparation of BiVO_4_ Electrode

The vanadium source solution was prepared by dissolving VO(acac)_2_ (0.5 M) in dimethylsulfoxide (10 mL). The as-prepared BiOI precursor film was dropped into the superfluous vanadium source solution. The electrode was then shifted to a muffle furnace and annealed for about 12 h. The heating rate was 3 °C/min to 120 °C, 0.67 °C/min to 280 °C, and 1.41 °C/min to 450 °C, and then held at 450 °C for 1 h. All the annealing processes ended with furnace cooling. After annealing, the electrodes were immersed in a 1.0 M NaOH solution for 15 min with gentle stirring to wash off the V_2_O_5_ on the BiVO_4_ surface. The prepared BiVO_4_ electrodes were rinsed thoroughly with deionized water and dried in a drying oven.

### 2.3. Photo-Assisted Electrodeposition of CoPi Cocatalyst

The CoPi cocatalyst was electrodeposited on the BiVO_4_ electrode under AM 1.5 G simulated sunlight by using the three-electrode system containing the solutions of NaH_2_PO_4_ (0.1 M), Na_2_HPO_4_ (0.1 M), Co(NO_3_)_2_·6H_2_O (0.001 M), and deionized water (100 mL). Similarly, an Ag/AgCl with a saturated KCl solution was used as the reference electrode and a Pt-foil as the counter electrode. The deposition voltage and time were −0.40 V_Ag/AgCl_ and 90 s, respectively.

### 2.4. Characterizations

The crystallinity and structure of BiVO_4_ films were examined by X-ray diffraction (XRD, Ultima-iv with Cu/K_α_ radiation). Surface morphologies were observed via scanning electron microscope (SEM, Germany Zeiss SUPRA 55). The transmittance and absorption of BiVO_4_ films were measured via a Shimadzu UV-3600 spectrophotometer. Ultraviolet photoelectron spectroscopy (UPS) was performed using a PHI 5000 VersaProbe with an energy value of a He I source with 21.22 eV. The PEC performance was obtained by using the CHI 660E electrochemical workstation under a three-electrode system, where an Ag/AgCl with saturated KCl solution was used as the reference electrode, with Pt-foil as the counter electrode and the BiVO_4_ photoanode as the working electrode. Photoelectrochemical impedance spectroscopy (PEIS) was assessed under light conditions, and its frequency ranged from 10^−1^ Hz to 10^5^ Hz. Mott–Schottky (M-S) measurements were used to calculate the flat band potential (*E_fb_*) of the BiVO_4_ films and the analysis of defects in junctions formed at the interface of semiconductor electrolytes.

## 3. Results and Discussion

BiVO_4_ film with different thicknesses can be prepared by adjusting the electrodeposition time of the BiOI precursor film, i.e., the BiOI-1 precursor film deposited at 300 s, BiOI-2 at 330 s, and BiOI-3 at 360 s, respectively, and then dropping excessive vanadium source solution to anneal in muffle furnace. The resulting BiVO_4_ films were labeled as BiVO_4_-1, BiVO_4_-2, and BiVO_4_-3, and the corresponding thicknesses were 358.5 nm, 424.3 nm, and 485.6 nm, respectively (Appendix A). The X-ray diffraction (XRD) patterns of BiVO_4_ film with three different thicknesses are shown in Figure 1a. The existence of three major diffraction peaks (011), (–121), and (040) and the standard monoclinic BiVO_4_ (JCPDS Card No.14-0688) peaks without any extra peaks, confirm the high crystallinity and pureness of the as-fabricated BiVO_4_ thin films [1]. Figure 1b shows that the smaller FWHM (full width at half maxima) values of different diffraction peaks demonstrate larger crystal grains in the BiVO_4_-2 film. The grain size can also be calculated according to the Scherrer formula [29]:(1)$$D=\frac{K\lambda }{\beta cos\theta }$$
where *K* is the Scherrer constant, *λ* is the wavelength of the X-ray sources (0.15406 nm), *D* is crystallite size (nm), *β* is the FWHM value, and *θ* is the Bragg angle at peak position. Figure 1c displays the proportion diagram of grain size distribution, and the BiVO_4_-2 obviously accounts for a large proportion in areas with a large grain size distribution. In addition, the average grain sizes of BiVO_4_-1, BiVO_4_-2, and BiVO_4_-3 are 15.54, 16.28, and 15.51 nm, respectively. The surface scanning electron microscopy (SEM) images of the BiVO_4_-1, BiVO_4_-2, and BiVO_4_-3 films are shown in Figure 1d–f. It can be seen that Figure 1e depicts larger crystalline grains compared to those of BiVO_4_-1 (Figure 1d). The BiVO_4_-1 film consists of many small grains with obvious voids on their surfaces, which are not conducive to charge transport, possibly due to the increase in charge recombination centers on the BiVO_4_-1 film surface. On the other hand, a quasi-uniform BiVO_4_-2 film with large grains can be obtained by increasing the deposition time of the BiOI precursor film, which is directly related to its effective thermodynamic/kinetic growth under a sufficient annealing temperature. However, the BiVO_4_-3 film shows a stacked structure and more voids than the BiVO_4_-2 film as the deposition time of the precursor film increases (Figure 1f). Therefore, highly compact and quasi-uniform BiVO_4_-2 films are better for photo-assisted electrodeposition with a CoPi catalyst.

The energy band gap (*E_g_*) of the BiVO_4_ semiconductor was gained from the transmission spectra in the wavelength range of 300 nm to 1500 nm, as shown in Figure 2a. The transmittance value of the BiVO_4_-2 film is lower than that of the BiVO_4_-1 and BiVO_4_-3 films, indicating that it has a higher absorbance. The FTO was used as a substrate, and the *E_g_* value was calculated using the equations below [30]:(2)$$\alpha =\ln\left(1/T\right)/d$$
(3)$$\alpha hv={C(hv-E_{g})}^{n}$$
where *n* is an index equal to 0.5 in a direct band-gap semiconductor, *d* is the thickness of the BiVO_4_ film, *T* is the transmission, *hν* is the energy of a photon, *α* is the absorption coefficient, and C is a constant. The BiVO_4_ films with three different thicknesses display a similar *E_g_* value of 2.41 eV (Figure 2b). The energy band information of the BiVO_4_ film was calculated via ultraviolet photoelectron spectroscopy (UPS), as we can see in Figure 2c. According to the secondary electron cut-off (SEC) edge and valence band (V_B_) position, the work function (Φ) of BiVO_4_ was calculated as 5.57 eV [31]; meanwhile, the E_F_ and E_V_ are determined as −5.57 eV and −7.23 eV, respectively (Appendix A). In combination with its optical band gap (*E_g_*), at a value of 2.41 eV, the conduction band (E_C_, vs. vacuum) of the BiVO_4_ semiconductor can be obtained, and its value is –4.82 eV.

The photocurrent density potential (J-V) curves of pure BiVO_4_ photoanodes with different thicknesses corresponding to the differing BiVO_4_/CoPi photoanodes are shown in Figure 2d. It can be seen that the BiVO_4_-2 photoanodes are significantly superior to other BiVO_4_ photoanodes, indicating that the BiVO_4_ film with the fewest charge recombination centers was obtained by adjusting the electrodeposition time of the BiOI precursor film and the annealing temperature. In addition, we can also see that the *J*_ph_ and the fill factor of all BiVO_4_ photoanodes increased after CoPi catalyst surface modification. Of these, the BiVO_4_-2/CoPi photoanode has a maximum *J*_ph_ value of 1.39 mA cm^−2^. Therefore, BiVO_4_-2 with optimal thickness was selected in this work in order to study the effect of CoPi catalyst on OER occurrence and PEC performance. The potential relative to the Ag/AgCl reference electrode (*V*_Ag/AgCl_) can be converted into *V_RHE_* using the Nernst equation [32]:(4)$$V_{RHE}=V_{Ag/AgCl}+0.059\times PH+0.198$$
while ABPE was obtained from the J-V response of the photoanodes according to the following equation [33]:(5)$$ABPE\left(\%\right)=J_{ph}\times \left(V_{RHE}-V_{H_{2}O/O_{2}}\right)/P_{sun}\times 100\%$$
where *V_RHE_* is the potential relative of a relative hydrogen electrode (*RHE*), $V_{H_{2}O/O_{2}}$ is the oxidation potential for oxygen (1.23 *V_RHE_*), and *P_sun_* is the simulated sunlight intensity (100 mW/cm^2^). The electrocatalytic OER activities of BiVO_4_ and BiVO_4_/CoPi electrodes were measured by linear sweep voltammetry (LSV) tests in 0.2 M Na_2_HPO_4_/NaH_2_PO_4_ solution (pH = 6.5) under dark conditions. Figure 3a depicts the LSV curves with a 100% iR drop compensation and a 0.1 mV s^−1^ scanning rate of the OER catalysts. In general, a catalyst’s OER activity is typically evaluated based on its overpotential at a current density of 10 mA cm^−2^ [27]. Therefore, the BiVO_4_/CoPi displays the overpotential value of 0.99 *V_RHE_*, which is smaller than that of BiVO_4_ (1.87 *V_RHE_*) at 10 mA cm^−2^ and the Tafel slope of BiVO_4_/CoPi (106.0 mV dec^−1^) is also obviously lower than that of BiVO_4_ (265.7 mV dec^−1^), indicating that the CoPi catalyst is able to drive reactions at lower overpotentials and play an excellent role in BiVO_4_ surface modification (Figure 3b,c). Moreover, the chopped J-V curves are relevant to photoanodes in the dark and continuous illumination within a lesser potential range (<1.5 *V_RHE_*) and are shown in Figure 3d. The dark current density is very low (~0 mA cm^−2^ at 1.23 *V_RHE_*); hence, under AM 1.5 G simulated sunlight illumination, the photocurrent density of BiVO_4_ and BiVO_4_/CoPi photoanodes increases sharply, confirming the rapid photo-generated carrier generation, separation, and transport, without the need for excess drive potentials. The J-V curves based on BiVO_4_ and BiVO_4_/CoPi photoanodes are displayed in Figure 3e. Specifically, the *J*_ph_ values at 1.23 *V_RHE_* are determined to be 0.75 mA cm^−2^ and 1.39 mA cm^−2^ in sequence. Furthermore, Figure 3f exhibits the as-calculated ABPE and the BiVO_4_/CoPi photoanode (0.45%) is about ~3 times higher than the BiVO_4_ photoanode (0.14%), indicating a simultaneous upgrade in PEC performance.

The charge transport kinetics were explored according to the electrochemical impedance test. Figure 4a,b exhibits the photoelectrochemical impedance spectroscopy (PEIS) characterization of the BiVO_4_ and BiVO_4_/CoPi photoanodes under illumination. The equivalent circuit is shown in Figure 4a; R_W_ can be attributed to the resistance of the electrolyte solution, and R_ct_ and C_sc_ represent the charge transport resistance and capacitance in the space charge region of the electrode/electrolyte interface. The fitted results are shown in Table 1, the chi-square values of the BiVO_4_ and BiVO_4_/CoPi photoanodes are both less than 0.02, and we can also see that the raw data and the fitted results match well (Figure 4a). Similar R_W_ values (18.2–18.8 Ω) indicate the stability of the test environment, and the remarkably smaller R_ct_ value of the BiVO_4_/CoPi photoanode (137.9 Ω) implies a more effective charge transport at the electrode/electrolyte interface. The larger C_sc_ value of the BiVO_4_/CoPi photoanode (5.20 × 10^−4^ F) indicates that the ability of the charge collection is strengthened. Generally, the PEIS-derived low-frequency region (10^−1^~10^1^ Hz) normally represents the mass transfer reactions in the electrode/electrolyte interface [34]. As shown in Figure 4b, the reduction of interface impedance (|Z|) also proves that the CoPi catalyst accelerates the mass transfer process. Moreover, the peaks in the Bode diagrams for the BiVO_4_ photoanode are located at low frequencies (10^−1^~10^1^ Hz), while the BiVO_4_/CoPi photoanode is located between 10^1^ and 10^2^ Hz (insert in Figure 4b). The rapid response of the BiVO_4_/CoPi photoanode to frequency indicates that charge transport and mass transfer simultaneously accelerated. The Mott–Schottky (M-S) measurement was also introduced in order to study the junction formed at the semiconductor–electrolyte interface in reaction to the applied potential (*V*_Ag/AgCl_, i.e., the aforementioned *E_appl_*). Figure 4c reveals that the 1/C^2^ increases with the potential *V*_Ag/AgCl_ in the presence of the space charge region (SCR), indicating n-type properties for the BiVO_4_ semiconductor. Moreover, the conduction band position of BiVO_4_ semiconductor relative to normal hydrogen electrode (*NHE*) can be obtained from the flat band potential (*E_fb_*), donor density (*N_D_*), the effective density of states functions in the conduction band (*N_c_*), and the effective mass of the electron ($m_{n}^{*}$), according to the following equations [35,36]:(6)$$\frac{1}{C_{sc}^{2}}=\frac{2}{eA^{2}\varepsilon \varepsilon_{0}N_{D}}(E_{appl}-E_{fb}-\frac{k_{B}T}{e})$$
(7)$$E_{c}(vs.NHE)=E_{fb}+kTln(\frac{N_{c}}{N_{D}})$$
(8)$$N_{c}=2(\frac{2\pi m_{n}^{*}kT}{h^{2}}{)}^{\frac{3}{2}}$$
where *C_sc_* is the SCR capacitance in the semiconductor–electrolyte interface, *A* is the active device area, *ε*_0_ is the permittivity of vacuum, *ε* is the relative dielectric coefficient, *k_B_* is Boltzmann constant, *e* is the unit charge, and *T* represents the temperature. The *E_fb_* value and *N_D_* value of the BiVO_4_ are 0.38 *V_NHE_* and 1.92 × 10^21^ cm^−3^, respectively. At room temperature (*T* = 25 °C), the vacuum energy level can be converted to *V_NHE_* by the following formula [37,38]:(9)$$Energy(vs.vacuum)={-eE}_{appl}\left(vs.NHE\right)-4.44\;eV$$

The calculated *E_C_* value range is –4.82 ± 0.02 eV, using the above equations, which is consistent with the *E_C_* of –4.82 eV obtained from the UPS measurement. According to the *E_g_* value (2.41 eV) calculated by transmission, the valance band (*E_V_*, vs. *NHE*) is 2.79 *V_NHE_*, which thermodynamically supports the occurrence of water oxidation with oxygen production by solar-driven water splitting. In contrast, the *E_fb_* value and N_D_ value of the BiVO_4_/CoPi quasi-semiconductor are 0.40 *V_NHE_* and 8.87 × 10^20^ cm^−3^, respectively (Figure 4d). The lower *N_D_* value demonstrates that the CoPi catalyst passivates defects (e.g., charge recombination centers) on the BiVO_4_ film surface, reducing electron–hole recombination during charge transport to the electrode/electrolyte interface, thereby improving photocurrent density. We can also clearly observe that the CoPi catalyst effectively reduces the defect pinholes on the BiVO_4_ surface through a top-view SEM image of the BiVO_4_/CoPi photoanode (Appendix A). The increase in *E_fb_* of 0.02 *V_NHE_* also indicates that the BiVO_4_/CoPi photoanode is more beneficial for water oxidation in thermodynamics (Appendix A).

The surface charge transfer efficiency (*η_tran_*) and bulk charge separation efficiency (*η_sep_*) of the BiVO_4_ photoanode were further investigated in order to determine the reasons for the significantly improved photocurrent density after surface modification with CoPi catalyst. The integrated photocurrent density (*J_abs_*) of BiVO_4_ and BiVO_4_/CoPi photoanodes can be obtained according to wavelength-dependent light harvesting efficiency (*LHE*) and the standard AM 1.5 G solar spectrum, utilizing the following formulas [1]:(10)$$J_{abs}=\int_{300}^{\lambda_{e}}\frac{\lambda }{1240}{\cdot N}_{ph}(\lambda )\cdot LHE(\lambda )d\lambda$$
(11)$$LHE=1-10^{-A(\lambda )}$$
where *J_abs_* is the integrated photocurrent density, *λ_e_* is the absorption cut-off wavelength that is linked to the band gap, *N_ph_* (*λ*) is the photo flux, and *A* (*λ*) is the wavelength-dependent absorption, covering wavelengths from 350 to 800 nm (Figure 5a). The *λ_e_* values for BiVO_4_ and BiVO_4_/CoPi were determined to be at 514 nm and 520 nm, suggesting CoPi can also effectively broaden and heighten the *LHE* range of the BiVO_4_ photoanode (Figure 5b), and giving the *J_abs_* values of 6.29 mA cm^−2^ and 6.68 mA cm^−2^, respectively (Figure 5c). Moreover, the transient photocurrent response spectra of the BiVO_4_ and BiVO_4_/CoPi photoanodes are shown in Figure 5d, while the *η_tran_* can be obtained through the measured photocurrents associated with the “light off” state and the “light on” state, according to the following formulas [32]:(12)$$\eta_{tran}=\frac{J_{ss}}{J_{inst}}$$
where *J_ss_* is the photocurrent density in steady state and *J_inst_* signifies the instantaneous photocurrent density. Accordingly, the *η_tran_* value of the BiVO_4_/CoPi photoanode (82.1%) has increased in comparison to the BiVO_4_ photoanode (76.0%), demonstrating that the CoPi catalyst accelerates holes transfer to the electrode/electrolyte interface and then oxidizes water. Moreover, we can observe in Figure 5d that the BiVO_4_/CoPi photoanode has a spike peak in the “light off” state compared with the BiVO_4_ photoanode. The negative current transient suggests that there is significant back electron/hole recombination after “light off” [39], reiterating that the BiVO_4_/CoPi photoanode has a larger capacitance value (Table 1), which demonstrates that the CoPi catalyst can delay charge recombination and promote the OER’s continuous progress. Additionally, the *η_sep_* can be calculated using following equation [32]:(13)$$\eta_{sep}=\frac{J_{ph}}{J_{abs}\times \eta_{tran}}$$

The BiVO_4_ and BiVO_4_/CoPi photoanodes’ calculated *η_sep_* values are 15.7%, and 25.4%, respectively. The obvious improvement of the *η_sep_* value indicates that the CoPi catalyst is also conducive to promoting the rapid separation of electron–hole pairs in the BiVO_4_ body.

## 4. Conclusions

In summary, a highly compact and quasi-uniform BiVO_4_ film was obtained at a suitable electrodeposition time and annealing temperature. After the successful electrodeposition of the CoPi catalyst on the optimized BiVO_4_ electrode, the FTO/BiVO_4_/CoPi photoanodes were fabricated, and their PEC performances were systematically investigated. Due to the surface modification of the CoPi catalyst, i.e., passivating charge recombination centers on the BiVO_4_ surface and promoting the separation of electron–hole pairs in the BiVO_4_ body, the interface impedance (|Z|) of the mass transfer process was decreased, while significantly enhancing the *η*_tran_ value of 82.1% and *η*_sep_ value of 25.4%. The *J*_ph_ and ABPE of the BiVO_4_ photoanode were increased by about ~2 times and ~3 times, respectively, demonstrating that the CoPi catalyst can accelerate holes transfer from the BiVO_4_ semiconductor to the catalyst in order to be competitive with water oxidation by holes in the semiconductor and to improve PEC performance. The results of this study may open up the possibility for the future design and construction of extremely effective photoanodes for solar-driven water splitting.

## Figures and Tables

**Figure 1 materials-16-03414-f001:**
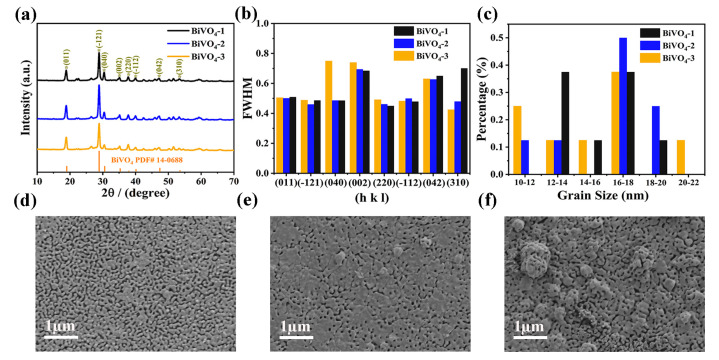
(**a**) XRD patterns; (**b**) full width at half maxima of different diffraction peaks; (**c**) grain size; and (**d**–**f**) surface SEM pictures of the BiVO_4_ films with different thicknesses.

**Figure 2 materials-16-03414-f002:**
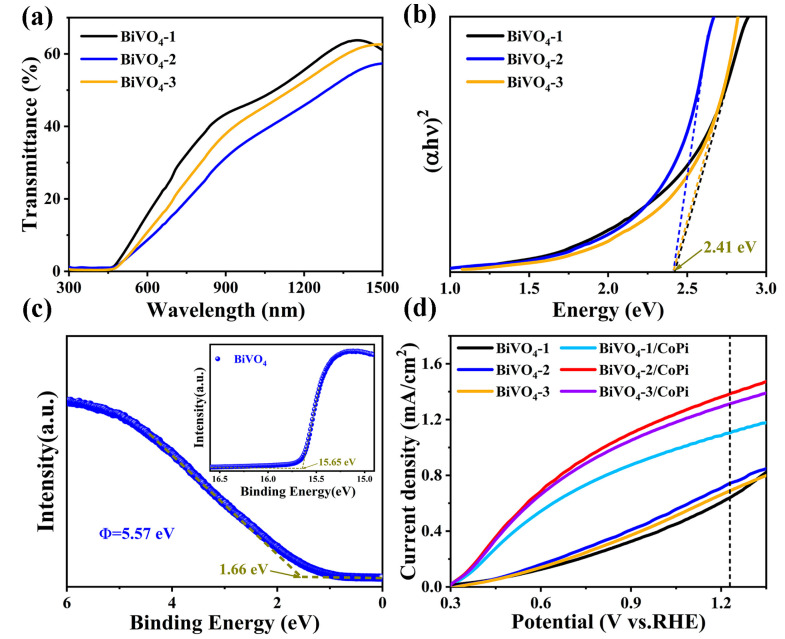
(**a**) Transmission spectra; (**b**) plots of (*αhv*)^2^ versus energy; (**c**) UPS characterizations deriving the SEC edge and the V_B_ position of BiVO_4_ film; and (**d**) J-V curves of the BiVO_4_ photoanodes with different thicknesses corresponding to differing BiVO_4_/CoPi photoanodes under sunlight illumination.

**Figure 3 materials-16-03414-f003:**
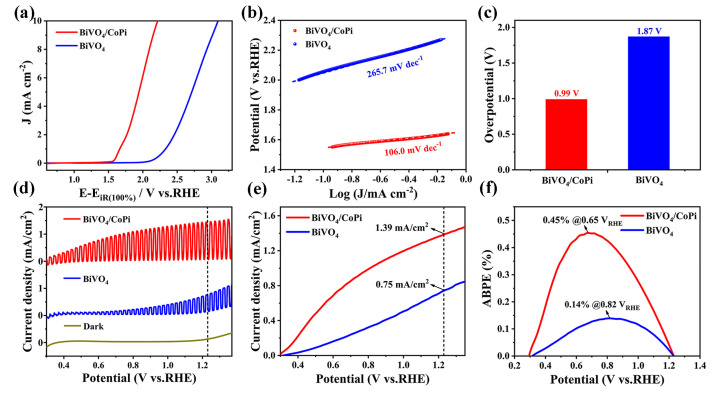
(**a**) LSV curves with 100% iR drop compensation and a 0.1 mV s^−1^ scanning rate, (**b**) Tafel plots, and (**c**) overpotential plots at a current density of 10 mA cm^−2^ of the electrochemical OER measurements with the BiVO_4_ and BiVO_4_/CoPi photoelectrode catalysts in 0.2 M Na_2_HPO_4_/NaH_2_PO_4_ solution (pH = 6.5) under dark conditions. (**d**) J-V curves of the BiVO_4_ and BiVO_4_/CoPi photoanodes under dark conditions and simulated sunlight irradiation; (**e**) J-V curves of the BiVO_4_ and BiVO_4_/CoPi photoanodes under sunlight illumination. (**f**) Calculated ABPE values.

**Figure 4 materials-16-03414-f004:**
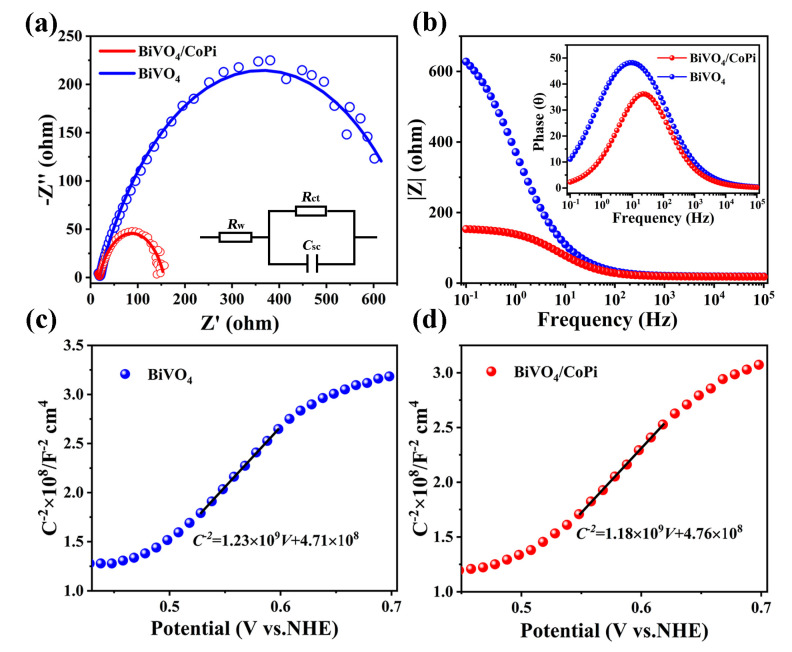
(**a**) Nyquist plots and equivalent circuit diagram (inset), where the small circles represent the raw data and the solid lines represent the fitting results, and (**b**) the corresponding Bode plots of the BiVO_4_ and BiVO_4_/CoPi photoanodes. M-S plots of (**c**) BiVO_4_ and (**d**) BiVO_4_/CoPi at a 1 kHz frequency.

**Figure 5 materials-16-03414-f005:**
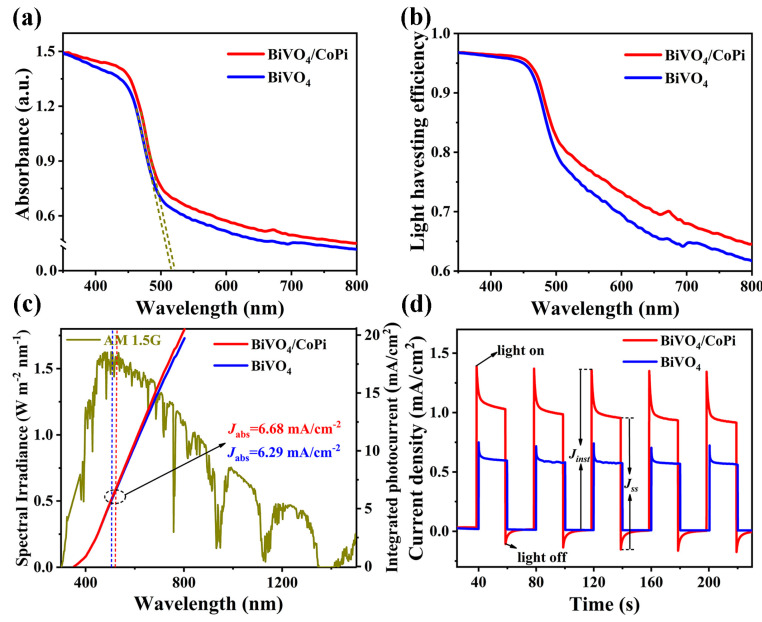
(**a**) Wavelength-dependent absorbance and (**b**) the light harvesting efficiency (*LHE*) of the BiVO_4_ and BiVO_4_/CoPi photoanodes. (**c**) The integrated photocurrent density of the BiVO_4_ and BiVO_4_/CoPi photoanodes, as well as the energy density flux for the standard AM 1.5 G solar spectrum. (**d**) Transient photocurrent response of the BiVO_4_ and BiVO_4_/CoPi photoanodes.

**Table 1 materials-16-03414-t001:** Summary of the PEIS fitted parameters for FTO/BiVO_4_ and FTO/BiVO_4_/CoPi photoanodes.

**Photoanodes**	***R*_W_ (Ω)**	***R*_ct_ (Ω)**	***C*_sc_ (F)**
FTO/BiVO_4_	18.8	690.3	5.03 × 10^−4^
FTO/BiVO_4_/CoPi	18.22	137.9	5.20 × 10^−4^

## Data Availability

The data presented in this study are available on request from the corresponding authors.

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
