# Peer review of "Charge Transport Enhancement in BiVO_4_ Photoanode for Efficient Solar Water Oxidation"

_materials, 2023, doi:10.3390/ma16093414_

Round 1
Reviewer 1 Report
In this manuscript (materials-2324972) the authors evaluated CoPi/BiVO4 and BiVO4 materials. Microstructural properties and photocurrent efficiency were obtained from XRD, SEM, UV-vis, UPS and photoelectrochemical and Electrochemical Impedance Spectroscopy (EIS) methods. Finally, CoPi/BiVO4 and BiVO4 materials were tested on Oxygen Evolution Reaction.
The writing and punctuation in English must be revised it. My recommendation is accepted with major amendment. Here, I give my recommendations.
Minor points
1. The abstract must be improved. I think the abstract don’t show the philosophy of the manuscript.
2. The conclusion must also be improved it.
3. In the EIS experiment, what is the quality of regression (Chi) for the equivalent circuit?
Major points
1. The authors performed the oxidation of water in Figure 3, why didn’t they consider the OER performance as the final application? I think the manuscript must be re-ordered it. For example, XRD, SEM, UV-vis, UPS and EIS methods could be the physicochemical characterization and the Electrochemical application applied to OER performance.
2. In the XRD analysis, do authors consider eliminating the contribution of FWHM from the equipment? The value of crystallite-average size could be overestimated; then the authors can perform Williamson-Hall method, as a first approximation. I suggest this reference “Estudillo‐Wong, Luis Alberto, and Nicolas Alonso‐Vante. "Factors Intervening in Oxide and Oxide‐Composite Supports on Nanocatalysts in the Energy Conversion." In Heterogeneous Nanocatalysis for Energy and Environmental Sustainability, John Wiley and Sons Ltd, 2022; pp 1-60” and which K-value did author consider to evaluate the crystallite-average size? The K-value close to 1 is applied for spherical morphology.
3. Why didn’t authors evaluate the grain-size from SEM images? The authors can compare with XRD analysis. Furthermore, authors suggested a nanorods morphology, in the BiVO4-2 material. However, with this SEM image alone, it is not possible to probe this statement.
4. The microstructural properties weren’t correlated with the photo-electrochemical test, why?
5. Authors said […] Generally, the PEIS-derived low-frequency region (10-1~101 Hz) normally represents the mass transfer reactions in the electrode/electrolyte interface […] The Nyquist plot didn’t show a mass transfer behavior and the equivalent circuit didn’t consider this phenomenon, why do authors affirm that phenomenon happened at lower frequencies?
6. How do authors calculate the bulk charge separation efficiency? They didn’t show anything.
7. Why is the ND value higher than 1018 carriers cm-3?
Some words are misspelled and some phrases like "The integrated photocurrent density (Jabs) of BiVO4 and BiVO4/CoPi photoanodes can be obtained according to wavelength-dependent light harvesting efficiency". The idea is clear but the sentence could be improved it.
Reviewer 2 Report
This manuscript presents a study on the electrochemical fabrication of FTO/BiVO4/CoPi photoanodes for the PEC applications. The article is fairly well written and relevant in the field. The authors have developed a strategy to enhance the charge transport ability of BiVO4 by surface modification with electrodeposited CoPi. This article could be considered for publication in ‘materials’ after minor corrections.
11. Even though electrodeposition is an important part in the manuscript and in the materials development process, there are only limited information about the electrodeposition experiments in the article. It would be nice to include cyclic voltammogram etc. in the manuscript or in ESI to support the selection of deposition potential for example.
22. In the electrodeposition experiments, it is better to compare the charge values instead of the time of deposition. Have you got the charge / charge density values corresponding to each sample?
33. Could you please explain how the initial 60s deposition at a higher negative potential (-0.4V) helps in the adhesion of the films?
44. How did you estimate the thickness of each film? It is better to show the results (ESI for eg.)
Minor grammar and spelling corrections required
Reviewer 3 Report
The submitted article is devoted to the synthesis and study of FTO/BiVO4/CoPi photoanodes for efficient solar water oxidation and is of interest to specialists in this field. However, I cannot recommend this article for publication as presented, as it has many weak points.
1. There are typos and errors in the text. The text style needs to be improved.
2. Your claim that Figure 1e exhibits many nanorods is debatable. It cannot be concluded from this figure that nanorods are present on the surface. TEM results should be presented.
3. Please explain in more detail how you obtained conduction band (EC, vs vacuum) (EC, vs vacuum) of BiVO4 equal –4.74 eV. There are no formulas and calculations indicating this value in the text.
4. Request to the author to give a detailed schematic diagram of the mechanism of this photoanode for a better understanding of the energy levels.
5. Please improve the description of equation 9.
6. How do the values of Efb and conduction bands EC, obtained by measuring the UPS.
English needs to be improved significantly.
For example, line 119: “The resulting BiVO4 film separately were labeled…”; line 121: “The X-ray diffraction (XRD) patterns of BiVO4 film with three different thicknesses was shown in Figure 1a” and so on.
Round 2
Reviewer 1 Report
In this manuscript (materials-2324972) the authors evaluated CoPi/BiVO4 and BiVO4 materials. Microstructural properties and photocurrent efficiency were obtained from XRD, SEM, UV-vis, UPS and photoelectrochemical and Electrochemical Impedance Spectroscopy (EIS) methods. Finally, CoPi/BiVO4 and BiVO4 materials were tested on Oxygen Evolution Reaction.
I just want to give my recommendations.
1. Authors didn't comment if the i-E profile was corrected by iR-drop in Figure 3(a), do they consider that?
2. In the EIS experiment, authors comment that the obtained chi-square was very lower. However, Figure 4(a) didn't show the fitting profile, I recommend to include it. Here, charge transport is more accelerated than mass transport at lower frequencies. The paper included suggest that phenomenon.
